# Signatures of the topological $s^{+-}$ superconducting order parameter in the type-II Weyl semimetal $T_d$-MoTe$_2$

Z. Guguchia[1], F. von Rohr[2], Z. Shermadini[3], A.T. Lee[4], S. Banerjee[4], A.R. Wieteska[1], C.A. Marianetti[4], B.A. Frandsen[5], H. Luetkens[3], Z. Gong[1], S.C. Cheung[1], C. Baines[3], A. Shengelaya[6,7], G. Taniashvili[6], A.N. Pasupathy[1], E. Morenzoni[3], S.J.L. Billinge [4,8], A. Amato [3], R.J. Cava[2], R. Khasanov[3] & Y.J. Uemura[1]

In its orthorhombic $T_d$ polymorph, MoTe$_2$ is a type-II Weyl semimetal, where the Weyl fermions emerge at the boundary between electron and hole pockets. Non-saturating magnetoresistance and superconductivity were also observed in $T_d$-MoTe$_2$. Understanding the superconductivity in $T_d$-MoTe$_2$, which was proposed to be topologically non-trivial, is of eminent interest. Here, we report high-pressure muon-spin rotation experiments probing the temperature-dependent magnetic penetration depth in $T_d$-MoTe$_2$. A substantial increase of the superfluid density and a linear scaling with the superconducting critical temperature $T_c$ is observed under pressure. Moreover, the superconducting order parameter in $T_d$-MoTe$_2$ is determined to have 2-gap $s$-wave symmetry. We also exclude time-reversal symmetry breaking in the superconducting state with zero-field μSR experiments. Considering the strong suppression of $T_c$ in MoTe$_2$ by disorder, we suggest that topologically non-trivial $s^{+-}$ state is more likely to be realized in MoTe$_2$ than the topologically trivial $s^{++}$ state.

[1] Department of Physics, Columbia University, New York, NY 10027, USA. [2] Department of Chemistry, Princeton University, Princeton, NJ 08544, USA. [3] Laboratory for Muon Spin Spectroscopy, Paul Scherrer Institute, CH-5232 Villigen PSI, Switzerland. [4] Department of Applied Physics and Applied Mathematics, Columbia University, New York, NY 10027, USA. [5] Department of Physics, University of California, Berkeley, CA 94720, USA. [6] Department of Physics, Tbilisi State University, Chavchavadze 3, GE-0128 Tbilisi, Georgia. [7] Andronikashvili Institute of Physics of I. Javakhishvili Tbilisi State University, Tamarashvili Str. 6, 0177 Tbilisi, Georgia. [8] Condensed Matter Physics and Materials Science Department, Brookhaven National Laboratory, Upton, NY 11973, USA. Correspondence and requests for materials should be addressed to Z.G. (email: zg2268@columbia.edu)

An interesting physical properties of two-dimensional materials such as transition metal dichalcogenides (TMDs) with a common formula, $MX_2$ (M is a transition metal, X is a chalcogen atom), are useful for many emerging technological applications[1–19]. Depending on the crystal structure, TMDs can be either semiconducting or semimetallic[20–23]. The title compound $MoTe_2$ undergoes a structural phase transition from monoclinic $1T'$ to orthorhombic $T_d$ at $T_S \sim 250\,\mathrm{K}$[15]. The $1T'$ structure possesses the inversion symmetric space group $P2_1/m$, whereas the $T_d$ phase belongs to the non-centrosymmetric space group $Pmn2_1$. Weyl fermions occur in the $T_d$ phase where the inversion symmetry is broken and $T_d$-$MoTe_2$ is considered to be type-II Weyl semimetal[1, 2]. The evidence for the low temperature $T_d$ structure in our $MoTe_2$ sample is provided by X-ray pair distribution function (PDF) measurements (Supplementary Note 1; Supplementary Figs. 3 and 4). The Fermi surfaces in a type-II Weyl semimetal consist of a pair of electron pockets and hole pockets touching at the Weyl node, rather than at the point-like Fermi surface in traditional type-I WSM systems. Well fermions can arise by breaking either the space-inversion (SIS) or time-reversal symmetry (TRS)[24–26]. The different symmetry classifications of the Weyl semimetals are expected to exhibit distinct topological properties. Recent angle-resolved photo-emission (ARPES) measurements[27] and a high-field quantum oscillation study[28] of the magnetoresistance (MR) in $T_d$-$MoTe_2$ revealed a distinctive features of surface states. In addition, in $Mo_xW_{1-x}Te_2$, experimental signatures of the predicted topological connection between the Weyl bulk states and Fermi arc surface states were also reported[29], constituting another unique property of Weyl semimetals.

$T_d$-$MoTe_2$ represents a rare example of a material with both superconductivity and a topologically non-trivial band structure. At ambient pressure, $T_d$-$MoTe_2$ is superconducting with $T_c \simeq 0.1\,\mathrm{K}$, but the application of a small pressure[15] or the substitution of S for Te[30] can markedly enhance $T_c$. $T_d$-$MoTe_2$ is believed to be a promising candidate for topological superconductivity (TSC) in a bulk material. TSCs are materials with unique electronic states consisting of a full pairing gap in the bulk and gapless surface states composed of Majorana fermions (MFs)[24–26]. In general, topological superfluidity and superconductivity are well-established phenomena in condensed matter systems. The A-phase of superfluid helium-3 constitutes an example of a charge neutral topological superfluid, whereas $Sr_2RuO_4$[31] is generally believed to be topological TRS-breaking superconductor. However, an example of a TRS invariant topological superconductor[24, 25] is thus far unprecedented, and $T_d$-$MoTe_2$ may be a candidate material for this category. Until now, the only known properties of the superconducting state in $T_d$-$MoTe_2$ are the pressure-dependent critical temperatures and fields[15]. Thus, a thorough exploration of superconductivity in $T_d$-$MoTe_2$ from both experimental and theoretical perspectives is required.

To further explore superconductivity and its possible topological nature in $T_d$-$MoTe_2$, it is critical to measure the superconducting order parameter of $T_d$-$MoTe_2$ on the microscopic level through measurements of the bulk properties. Thus, we concentrate on high pressure[32–35] muon-spin relaxation/rotation (μSR) measurements of the magnetic penetration depth $\lambda$ in $T_d$-$MoTe_2$. This quantity is one of the fundamental parameters of a superconductor, as it is related to the superfluid density $n_s$ via $1/\lambda^2 = \mu_0 e^2 n_s/m^\star$ (where $m^\star$ is the effective mass). Remarkably, the temperature dependence of $\lambda$ is particularly sensitive to the topology of the SC gap: whereas in a nodeless superconductor, $\Delta\lambda^{-2}(T) \equiv \lambda^{-2}(0) - \lambda^{-2}(T)$ vanishes exponentially at low $T$, in a nodal SC it vanishes as a power of $T$. The μSR technique provides a powerful tool to measure $\lambda$ in the vortex state of type-II superconductors in the bulk of the sample, in contrast to many

techniques that probe $\lambda$ only near the surface[36]. Details are provided in the "Methods" section. In addition, zero-field μSR has the ability to detect internal magnetic fields as small as 0.1 G without applying external magnetic fields, making it a highly valuable tool for probing spontaneous magnetic fields due to TRS breaking in exotic superconductors.

By combining high-pressure μSR and AC-susceptibility experiments, we observed a substantial increase of the superfluid density $n_s/m^\star$ and a linear scaling with $T_c$ under pressure. Moreover, the superconducting order parameter in $T_d$-$MoTe_2$ is determined to have 2-gap $s$-wave symmetry. We also excluded time-reversal symmetry breaking in the high-pressure SC state, classifying $MoTe_2$ as time-reversal-invariant superconductor with broken inversion symmetry. Taking into account the previous report on the strong suppression of $T_c$ in $MoTe_2$ by disorder, we suggest that topologically non-trivial $s^{+-}$ state is more likely to be realized in $MoTe_2$ than the topologically trivial $s^{++}$ state. Should $s^{+-}$ indeed be the SC gap symmetry, the $T_d$-$MoTe_2$ is, to our knowledge, the first known example of a time-reversal-invariant topological (Weyl) superconductor.

## Results

**Probing the vortex state as a function of pressure.** Figure 1a shows the temperature dependence of the AC-susceptibility $\chi_{AC}$ of $T_d$-$MoTe_2$ in the temperature range between 1.4 and 4.2 K for

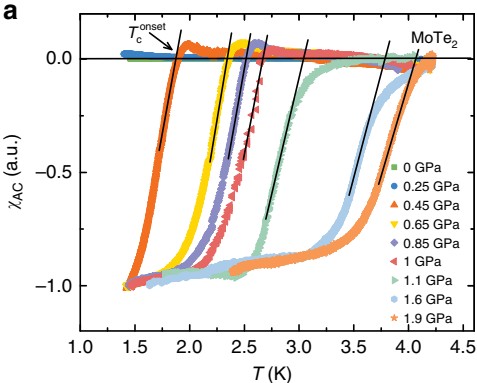

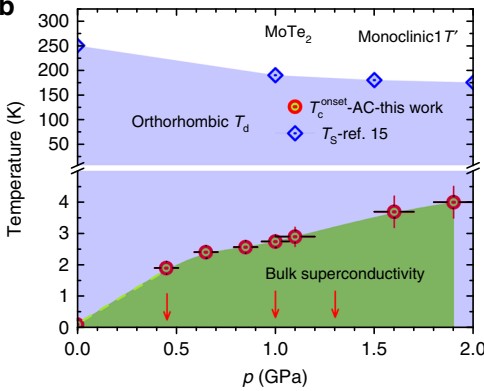

**Fig. 1** AC-susceptibility as a function of temperature and pressure in $MoTe_2$. **a** Temperature dependence of the AC-susceptibility $\chi_{AC}$ for the polycrystalline sample of $MoTe_2$, measured at ambient and various applied hydrostatic pressures up to $p \simeq 1\,\mathrm{GPa}$. The arrow denotes the superconducting transition temperature $T_c$. **b** Pressure dependence of $T_c$ (this work) and the structural phase transition temperature $T_S$[15]. Arrows mark the pressures at which the $T$-dependence of the penetration depth was measured

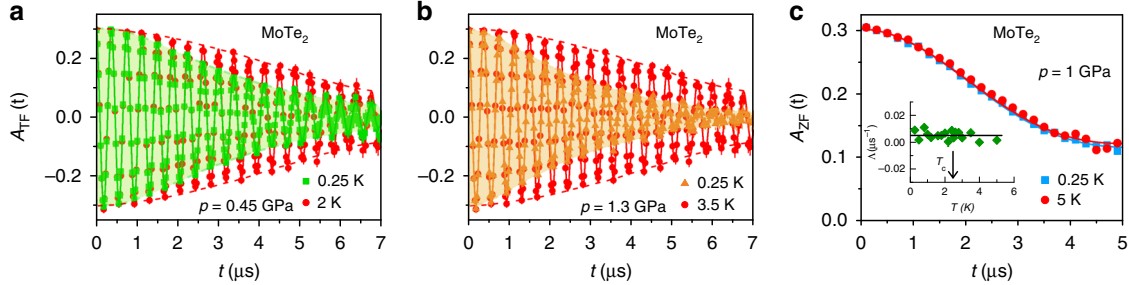

**Fig. 2** Transverse-field (TF) and zero-field (ZF) μSR-time spectra for MoTe$_2$. The TF spectra are obtained above and below $T_c$ in an applied magnetic field of $\mu_0 H = 20$ mT (after field cooling the sample from above $T_c$) at $p = 0.45$ GPa (**a**) and $p = 1.3$ GPa (**b**). The solid lines in **a** and **b** represent fits to the data by means of Eq. (1). The dashed lines are guides to the eye. **c** ZF μSR time spectra for MoTe$_2$ recorded above and below $T_c$. The line represents the fit to the data with a Kubo–Toyabe depolarization function[39], reflecting the field distribution at the muon site created by the nuclear moments. Error bars are the s.e. m. in about $10^6$ events. The error of each bin count $n$ is given by the s.d. of $n$. The errors of each bin in $A(t)$ are then calculated by s.e. propagation

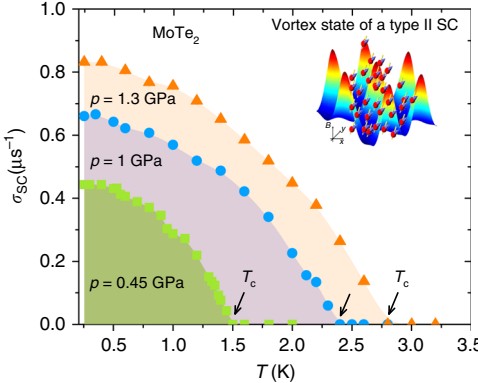

**Fig. 3** Superconducting muon-spin depolarization rate for MoTe$_2$. The colored symbols represent the depolarization rate $\sigma_{sc}(T)$ measured in an applied magnetic field of $\mu_0 H = 20$ mT at various temperatures and hydrostatic pressures. The arrows mark the $T_c$ values. Inset illustrates how muons, as local probes, sense the inhomogeneous field distribution in the vortex state of type-II superconductor. The error bars represent the s.d. of the fit parameters

selected hydrostatic pressures up to $p = 1.9$ GPa. A strong dia-magnetic response and sharp SC transition are observed under pressure (Fig. 1), pointing to the high quality of the sample and providing evidence for bulk superconductivity in MoTe$_2$[15]. The pressure dependence of $T_c$ is shown in Fig. 1b. $T_c$ increases with increasing pressure and reaches a critical temperature $T_c \simeq 4$ K at $p = 1.9$ GPa, the maximum applied pressure in the susceptibility experiments. The substantial increase of $T_c$ from $T_c \simeq 0.1$ K at ambient pressure to $T_c \simeq 4$ K at moderate pressures in MoTe$_2$ was considered as a manifestation of its topologically non-trivial electronic structure. Note that a strong pressure-induced enhancement of $T_c$ has also been observed in topological super-conductors such as Bi$_2$Te$_3$[37] and Bi$_2$Se$_3$[38]. The temperature of the structural phase transition from monoclinic 1T' to orthorhombic $T_d$[15] as a function of pressure is also shown in Fig. 1b. In the temperature and pressure range ($p = 0$–1.9 GPa) investigated here, MoTe$_2$ is in the orthorhombic $T_d$ structure. Moreover, density functional theory (DFT) calculations confirmed that in the pressure range investigated in this work, MoTe$_2$ is a Weyl semimetal in which the band structure near the Fermi level is highly sensitive to changes in the lattice constants[15].

Figure 2a and b displays the transverse-field (TF) μSR-time spectra for MoTe$_2$ measured at $p = 0.45$ GPa and the maximum applied pressure $p = 1.3$ GPa, respectively, in an applied magnetic field of $\mu_0 H = 20$ mT. Spectra collected above the SC transition temperature (2 K, 3.5 K) and below it (0.25 K) are shown. The presence of the randomly oriented nuclear moments causes a weak relaxation of the μSR signal above $T_c$. The relaxation rate is strongly enhanced below $T_c$, which is caused by the formation of a flux-line lattice (FLL) in the SC state, giving rise to an inhomogeneous magnetic field distribution. Another reason for an enhancement of the relaxation rate could be magnetism, if present in the samples. However, precise zero-field (ZF)-μSR experiments does not show any indication of magnetism in $T_d$-MoTe$_2$ down to 0.25 K. This can be seen in ZF time spectra, shown in Fig. 2c, which can be well described only by considering the field distribution created by the nuclear moments[39]. More-over, no change in ZF-μSR relaxation rate (see the inset of Fig. 2c) across $T_c$ was observed, pointing to the absence of any spontaneous magnetic fields associated with a TRS[31, 40, 41] breaking pairing state in MoTe$_2$.

Figure 3 displays the temperature dependence of the muon-spin depolarization rate $\sigma_{sc}$ (measured in an applied magnetic field of $\mu_0 H = 20$ mT) in the SC state of MoTe$_2$ at selected pressures. This relaxation rate is proportional to the width of the non-uniform field distribution (see "Methods" section). The formation of the vortex lattice below $T_c$ causes an increase of the relaxation rate $\sigma_{sc}$. As the pressure is increased, both the low-temperature value of $\sigma_{sc}(0.25$ K) and the transition temperature $T_c$ show a substantial increase (Fig. 3). $\sigma_{sc}(0.25$ K) increases by a factor of ~2 from $p = 0$ GPa to $p = 1.3$ GPa. In the following, we show that the observed temperature dependence of $\sigma_{sc}$, which reflects the topology of the SC gap, is consistent with the presence of the two isotropic s-wave gaps on the Fermi surface of MoTe$_2$.

**Pressure-dependent magnetic penetration depth.** To explore the symmetry of the SC gap, it is important to note that $\lambda(T)$ is related to $\sigma_{sc}(T)$ as follows[42]:

$$\frac{\sigma_{sc}(T)}{\gamma_\mu} = 0.06091 \frac{\Phi_0}{\lambda_{eff}^2(T)},\qquad(1)$$

where $\Phi_0$ is the magnetic-flux quantum and $\gamma_\mu$ denotes the gyromagnetic ratio of the muon. Thus, the flat $T$-dependence of $\sigma_{sc}$ at low temperature observed at various pressures (Fig. 3) implies an isotropic superconducting gap. In this case, $\lambda_{eff}^{-2}(T)$ exponentially approaches its zero-temperature value. We note that it is the effective penetration depth $\lambda_{eff}$ (powder average), which we extract from the μSR depolarization rate (Eq. (1)), and

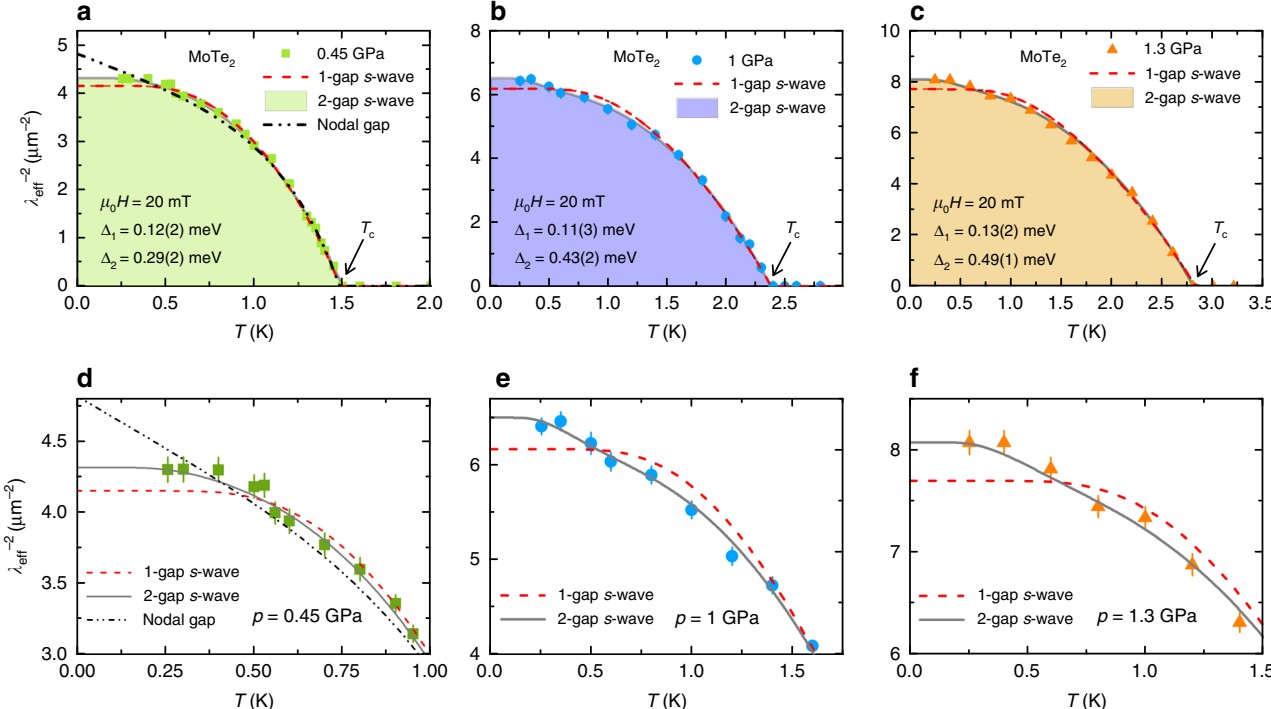

**Fig. 4** Pressure evolution of the penetration depth for MoTe$_2$. Colored symbols represent the value of $\lambda_{eff}^{-2}$ as a function of temperature, measured in an applied magnetic field of $\mu_0 H = 20$ mT under the applied hydrostatic pressures indicated in each panel. The solid lines correspond to a 2-gap s-wave model, the dashed and the dotted lines represent a fit using a 1-gap s-wave and nodal gap models, respectively. The error bars are calculated as the s.e.m

this is the one shown in the figures. In polycrystalline samples of highly anisotropic systems $\lambda_{eff}$ is dominated by the shorter penetration depth $\lambda_{ab}$ and $\lambda_{eff} = 1.3\lambda_{ab}$ as previously shown[43, 44].

The temperature dependence of the penetration depth is quantitatively described within the London approximation ($\lambda \gg \xi$, where $\xi$ is the coherence length) and by using the the empirical $\alpha$-model. This model[45–49] assumes, besides common $T_c$, that the gaps in different bands are independent of each other. The superfluid densities, calculated for each component independently[49], (see details in the "Methods" section) are added together with a weighting factor:

$$\frac{\lambda^{-2}(T)}{\lambda^{-2}(0)} = \alpha \frac{\lambda_{eff}^{-2}(T, \Delta_{0,1})}{\lambda_{eff}^{-2}(0, \Delta_{0,1})} + (1-\alpha) \frac{\lambda_{eff}^{-2}(T, \Delta_{0,2})}{\lambda_{eff}^{-2}(0, \Delta_{0,2})}, \quad (2)$$

where $\lambda_{eff}(0)$ is the effective penetration depth at zero temperature, $\Delta_{0,i}$ is the value of the $i$-th SC gap ($i = 1, 2$) at $T = 0$ K, $\alpha$ and $(1-\alpha)$ are the weighting factors, which measure their relative contributions to $\lambda^{-2}$.

The results of this analysis are presented in Fig. 4a–f, where the temperature dependence of $\lambda_{eff}^{-2}$ for MoTe$_2$ is plotted at various pressures. We consider two different possibilities for the gap function: either a constant gap, $\Delta_{0,i} = \Delta_i$, or an angle-dependent gap of the form $\Delta_{0,i} = \Delta_i \cos 2\varphi$, where $\varphi$ is the polar angle around the Fermi surface. The dashed and the solid lines represent fits to the data using a 1-gap s-wave and a 2-gap s-wave model, respectively. The analysis appears to rule out the simple 1-gap s-wave model as an adequate description of $\lambda_{eff}^{-2}(T)$ for MoTe$_2$. The 2-gap s-wave scenario with a small gap $\Delta_1 \simeq 0.12(3)$ meV and a large gap $\Delta_2$ (with the pressure-independent weighting factor of $1-\alpha = 0.87$), describes the experimental data remarkably well. The possibility of a nodal gap was also tested, shown with a black dotted line in Fig. 4a, but was found to be inconsistent with the data. This conclusion is supported by a $\chi^2$ test, revealing a value of $\chi^2$ for the nodal gap model that is ~30% higher than the one for

2-gap s-wave model for $p = 0.45$ GPa. The ratios of the SC gap to $T_c$ at $p = 0.45$ GPa were estimated to be $2\Delta_1/k_B T_c = 1.5(4)$ and $2\Delta_2/k_B T_c = 4.6(5)$ for the small and the large gaps, respectively. The ratio for the higher gap is consistent with the strong coupling limit BCS expectation[50]. However, a similar ratio can also be expected for Bose Einstein condensation (BEC)-like picture as pointed out in ref. [51]. It is important to note that the ratio $2\Delta/k_B T_c$ does not effectively distinguish between BCS or BEC. This is particularly true in two band systems, where the ratio is not universal even in the BCS limit, as it depends also on the density of states of the two bands. The pressure dependence of various physical parameters are plotted in Fig. 5a and b. From Fig. 5a, a substantial decrease of $\lambda_{eff}(0)$ (increase of $\sigma_{sc}$) with pressure is evident. At the highest applied pressure of $p = 1.3$ GPa, the reduction of $\lambda_{eff}(0)$ is ~25% compared with the value at $p = 0.45$ GPa. The small gap $\Delta_1 \simeq 0.12(3)$ meV stays nearly unchanged by pressure, whereas the large gap $\Delta_2$ increases from $\Delta_2 \simeq 0.29(1)$ meV at $p = 0.45$ GPa to $\Delta_2 \simeq 0.49(1)$ meV at $p = 1.3$ GPa, i.e., by ~70%.

In general, the penetration depth $\lambda$ is given as a function of $n_s$, $m^\star$, $\xi$, and the mean free path $l$ as

$$\frac{1}{\lambda^2} = \frac{4\pi n_s e^2}{m^* c^2} \times \frac{1}{1+\xi/l}. \quad (3)$$

For systems close to the clean limit, $\xi/l \to 0$, the second term essentially becomes unity, and the simple relation $1/\lambda \propto n_s/m^\star$ holds. Considering the $H_{c2}$ values of MoTe$_2$ reported in ref. [15], we estimated $\xi \simeq 26$ and 14 nm for $p = 0.45$ and 1 GPa, respectively. At ambient pressure, the in-plane mean free path $l$ was estimated to be $l \simeq 100$–200 nm[28]. No estimates are currently available for $l$ under pressure. However, in-plane $l$ is most probably independent of pressure, considering the fact that the effect of compression is mostly between layers rather than within layers, thanks to the unique anisotropy of the van der Waals structure. In particular, the intralayer Mo–Te bond length is almost unchanged

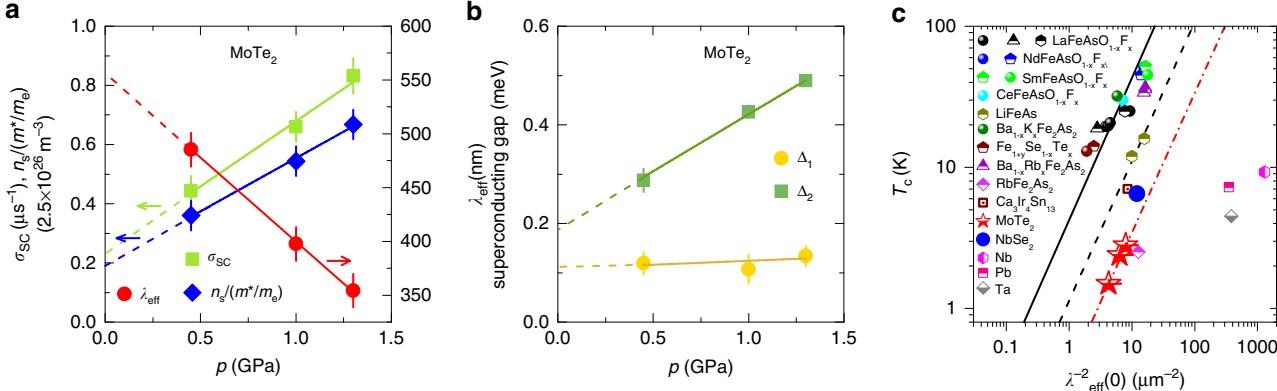

**Fig. 5** Pressure evolution of various quantities. The SC muon depolarization rate $\sigma_{SC}$, magnetic penetration depth $\lambda_{eff}$ and the superfluid density $n_s/m^*m_e$ (**a**) as well as the zero-temperature gap values $\Delta_{1,2}(0)$ (**b**) are shown as a function of hydrostatic pressure. Dashed lines are guides to the eye and solid lines represent linear fits to the data. The error bars represent the s.d. of the fit parameters. **c** A plot of $T_c$ vs. $\lambda_{eff}^{-2}(0)$ obtained from our μSR experiments in MoTe$_2$. The dashed red line represents the linear fit to the MoTe$_2$ data. The Uemura plot for various cuprate and Fe-based HTSs is also shown[49, 66–70]. The relation observed for underdoped cuprates is also shown (solid line for hole doping[55-59] and dashed black line for electron doping[61]). The points for various conventional BCS superconductors and for NbSe$_2$ are also shown

by pressure, especially in the pressure region relevant to this study. Thus, in view of the short coherence length and relatively large $l$, we can assume that MoTe$_2$ lies close to the clean limit[52]. With this assumption, we obtain the ground-state value $n_s/(m^*/m_e) \simeq 0.9 \times 10^{26}$ m$^{-3}$, $1.36 \times 10^{26}$ m$^{-3}$, and $1.67 \times 10^{26}$ m$^{-3}$ for $p = 0.45$, 1, and 1.3 GPa respectively. Interestingly, $n_s/(m^*/m_e)$ increases substantially under pressure, which will be discussed below.

## Discussion

One of the essential findings of this paper is the observation of two-gap superconductivity in $T_d$-MoTe$_2$. Recent ARPES[27] experiments on MoTe$_2$ revealed the presence of three bulk hole pockets (a circular hole pocket around the Brillouin zone center and two butterfly-like hole pockets) and two bulk electron pockets, which are symmetrically distributed along the Γ-X direction with respect to the Brillouin zone center Γ. As several bands cross the Fermi surface in MoTe$_2$, two-gap super-conductivity can be understood by assuming that the SC gaps open at two distinct types of bands. Now the interesting question arises: How consistent is the observed two-gap superconductivity with the possible topological nature of superconductivity in $T_d$-MoTe$_2$? Note that the superconductor $T_d$-MoTe$_2$ represents a time-reversal-invariant Weyl semimetal, which has broken inversion symmetry. Recently, the detailed studies of microscopic interactions and the SC gap symmetry for time-reversal-invariant TSC in Weyl semimetals were performed[24]. Namely, it was shown that for TSC the gaps can be momentum independent on each FS but must change the sign between different FSs. μSR experiments alone cannot distinguish between sing-changing $s^{+-}$ (topological) and $s^{++}$ (trivial) pairing states. However, considering the recent experimental observations of the strong suppression of $T_c$ in MoTe$_2$ by disorder[11, 53] and the theoretical proposal that TSC is more sensitive to disorder than the ordinary $s$-wave superconductivity[24, 54], we suggest that $s^{+-}$ state is more likely to be realized than the trivial $s^{++}$ state. Further phase sensitive experiments are desirable to distinguish between $s^{+-}$ and $s^{++}$ states in MoTe$_2$.

Besides the two-gap superconductivity, another interesting observation is the strong enhancement of the superfluid density $\lambda_{eff}^{-2}(0) \propto n_s/(m^*/m_e)$ and its linear scaling with $T_c$ (Fig. 5c). Between $p = 0.45$ and 1.3 GPa, $n_s/(m^*/m_e)$ increases by factor of

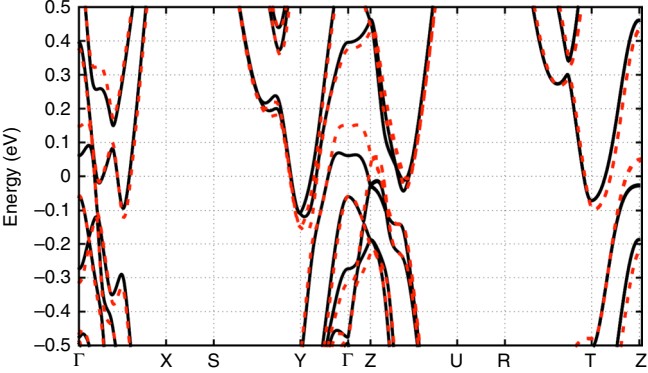

**Fig. 6** DFT results. Calculated band structure of $T_d$-MoTe$_2$ at ambient $p$ (solid black curves) and for $p = 1.3$ GPa (dashed red curves)

~1.8. We also compared the band structures for ambient as well as for the hydrostatic pressure of 1.3 GPa by means of DFT cal-culations. The results are shown in Fig. 6. When the pressure is applied, there are appreciable differences of the bands near the Fermi level, especially near $Y - Z$, $T - Z$, and $\Gamma - X$. Near Γ, the hole band is shifted by +0.8–0.9 eV, whereas the electron band at $Y$ and $T$ are lowered by 20–40 meV.

The nearly linear relationship between $T_c$ and the superfluid density was first noticed in hole-doped cuprates in 1988–1989[55, 56], and its possible relevance to the crossover from BEC to BCS condensation has been discussed in several sub-sequent papers[57–59]. The linear relationship was noticed mainly in systems lying along the line for which the ratio of $T_c$ to the effective Fermi temperature $T_F$ is about $T_c/T_F \sim 0.05$, implying a reduction of $T_c$ by a factor of 4–5 from the ideal Bose con-densation temperature for a non-interacting Bose gas composed of the same number of Fermions pairing without changing their effective masses. The present results on MoTe$_2$ and NbSe$_2$[60] in Fig. 5c demonstrate that a linear relation holds for these systems, but with the ratio $T_c/T_F$ being reduced by a factor of 16–20. It was also noticed[61] that electron-doped cuprates follow another line with their $T_c/T_F$ reduced by a factor of ~4 from the line of hole-doped cuprates. As the present system MoTe$_2$ and NbSe$_2$ fall into the clean limit, the linear relation is unrelated to pair breaking, and can be expected to hold between $T_c$ and $n_s/m^*$.

In a naive picture of BEC to BCS crossover, systems with small $T_c/T_F$ (large $T_F$) are considered to be on the "BCS" side, whereas the linear relationship between $T_c$ and $T_F$ is expected only on the BEC side. Figure 5c indicates that the BEC-like linear relationship may exist in systems with $T_c/T_F$ reduced by a factor 4 to 20 from the ratio in hole-doped cuprates, presenting a new challenge for theoretical explanations.

In conclusion, we provide the first microscopic investigation of the superconductivity in $T_d$-MoTe$_2$. Specifically, the zero-temperature magnetic penetration depth $\lambda_{eff}(0)$ and the temperature dependence of $\lambda_{eff}^{-2}$ were studied in the type-II Weyl semimetal $T_d$-MoTe$_2$ by means of µSR experiments as a function of pressure up to $p \simeq 1.3$ GPa. Remarkably, the temperature dependence of $1/\lambda_{eff}^2(T)$ is inconsistent with a simple isotropic $s$-wave pairing symmetry and with presence of nodes in the gap. However, it is well described by a 2-gap $s$-wave scenario, indicating multigap superconductivity in MoTe$_2$. We also excluded time-reversal symmetry breaking in the high-pressure SC state with sensitive zero-field µSR experiments, classifying MoTe$_2$ as time-reversal-invariant superconductor with broken inversion symmetry. In this type of superconductor, a 2-gap $s$-wave model is consistent with a topologically non-trivial superconducting state if the gaps $\Delta_1$ and $\Delta_2$ existing on different Fermi surfaces have opposite signs. µSR experiments alone cannot distinguish between sign changing $s^{+-}$ (topological) and $s^{++}$ (trivial) pairing states. However, considering the previous report on the strong suppression of $T_c$ in MoTe$_2$ by disorder, we suggest that $s^{+-}$ state is more likely to be realized in MoTe$_2$ than the $s^{++}$ state. Should $s^{+-}$ be the SC gap symmetry, the high-pressure state of MoTe$_2$ is, to our knowledge, the first known example of a Weyl superconductor, as well as the first example of a time-reversal invariant topological (Weyl) superconductor. Finally, we observed a linear correlation between $T_c$ and the zero-temperature superfluid density $\lambda_{eff}^{-2}(0)$ in MoTe$_2$, which together with the observed two-gap behavior, points to the unconventional nature of superconductivity in $T_d$-MoTe$_2$. We hope the present results will stimulate theoretical investigations to obtain a microscopic understanding of the relation between superconductivity and the topologically non-trivial electronic structure of $T_d$-MoTe$_2$.

## Methods

**Sample preparation.** High quality single crystals and polycrystalline samples were obtained by mixing of molybdenum foil (99.95%) and tellurium lumps (99.999+%) in a ratio of 1:20 in a quartz tube and sealed under vacuum. The reagents were heated to 1000 °C within 10 h. They dwelled at this temperature for 24 h, before they were cooled to 900 °C within 30 h (polycrystalline sample) or 100 h (single crystals). At 900 °C the tellurium flux was spined-off and the samples were quenched in air. The obtained MoTe$_2$ samples were annealed at 400 °C for 12 h to remove any residual tellurium.

**Pressure cell.** Single wall CuBe piston-cylinder type of pressure cell is used together with Daphne oil to generate hydrostatic pressures for µSR experiments[32, 33]. Pressure dependence of the SC critical temperature of tiny indium piece is used to measure the pressure. The fraction of the muons stopping in the sample was estimated to be ~40%.

**µSR experiment.** Nearly perfectly spin-polarized, positively charged muons $\mu^+$ are implanted into the specimen, where they behave as very sensitive microscopic magnetic probes. Muon-spin experiences the Larmor precession either in the local field or in an applied magnetic field. Fundamental parameters such as the magnetic penetration depth $\lambda$ and the coherence length $\xi$ can be measured in the bulk of a superconductor by means of transverse-field µSR technique, in which the magnetic field is applied perpendicular to the initial muon-spin polarization. If a type-II superconductor is cooled below $T_c$ in an applied magnetic field ranged between the lower ($H_{c1}$) and the upper ($H_{c2}$) critical fields, a flux-line lattice is formed and muons will randomly probe the non-uniform field distribution of the vortex lattice.

Combination of high-pressure µSR instrument GPD ($\mu$E1 beamline), the low-background instrument GPS ($\pi$M3 beamline) and the low-temperature instrument LTF ($\pi$M3.3) of the Paul Scherrer Institute (Villigen, Switzerland) is used to study the single crystalline as well as the polycrystalline samples of MoTe$_2$.

**Analysis of TF-µSR data.** The following function is used to analyze the TF µSR data[45]:

$$P(t) = A_s \exp\left[-\frac{(\sigma_{sc}^2 + \sigma_{nm}^2)t^2}{2}\right]\cos(\gamma_\mu B_{int,s}t + \varphi)$$
$$+ A_{pc}\exp\left[-\frac{\sigma_{pc}^2 t^2}{2}\right]\cos(\gamma_\mu B_{int,pc}t + \varphi) \qquad (4)$$

Here $A_s$ and $A_{pc}$ denote the initial assymmetries of the sample and the pressure cell, respectively. $\gamma/(2\pi) \simeq 135.5$ MHz/T is the gyromagnetic ratio of muon and $\varphi$ denotes the initial phase of the muon-spin ensemble. $B_{int}$ represents the internal magnetic field, sensed by the muons. $\sigma_{nm}$ is the relaxation rate, caused by the nuclear magnetic moments. The value of $\sigma_{nm}$ was obtained above $T_c$ and was kept constant over the entire temperature range. The relaxation rate $\sigma_{sc}$ describes the damping of the µSR signal due to the formation of the vortex lattice in the SC state. $\sigma_{pc}$ describes the depolarization due to the nuclear moments of the pressure cell. $\sigma_{pc}$ exhibits the temperature dependence below $T_c$ due to the influence of the diamagnetic moment of the SC sample on the pressure cell[34]. The linear coupling between $\sigma_{pc}$ and the field shift of the internal magnetic field in the SC state was assumed to consider the temperature-dependent $\sigma_{pc}$ below $T_c$: $\sigma_{pc}(T) = \sigma_{pc}(T > T_c)$ + $C(T)(\mu_0 H_{int,NS} - \mu_0 H_{int,SC})$, where $\sigma_{pc}(T > T_c) = 0.25$ µs$^{-1}$ is the temperature-independent Gaussian relaxation rate. $\mu_0 H_{int,NS}$ and $\mu_0 H_{int,SC}$ are the internal magnetic fields measured in the normal and in the SC state, respectively. As demonstrated by the solid lines in Fig. 2b and c, the µSR data are well described by Eq. (1).

**Analysis of $\lambda(T)$.** $\lambda_{eff}(T)$ was calculated by considering the London approximation ($\lambda \gg \xi$) using the following function[45, 46]:

$$\frac{\lambda_{eff}^{-2}(T, \Delta_{0,i})}{\lambda_{eff}^{-2}(0, \Delta_{0,i})} = 1 + \frac{1}{\pi}\int_0^{2\pi}\int_{\Delta_{(T,\varphi)}}^{\infty}\left(\frac{\partial f}{\partial E}\right)\frac{E dE d\varphi}{\sqrt{E^2 - \Delta_i(T,\varphi)^2}}, \qquad (5)$$

where $f = [1 + \exp(E/k_B T)]^{-1}$ represents the Fermi function, $\varphi$ is the angle along the Fermi surface, and $\Delta_i(T, \varphi) = \Delta_{0,i}\Gamma(T/T_c)g(\varphi)$ ($\Delta_{0,i}$ is the maximum gap value at $T$ = 0). The temperature evolution of the gap is given by the expression $\Gamma(T/T_c) =$ tanh$\{1.82[1.018(T_c/T - 1)]^{0.51}\}$[47], whereas $g(\varphi)$ takes care of the angular dependence of the superconducting gap. Namely, $g(\varphi) = 1$ in the case of both a 1-gap $s$-wave and a 2-gap $s$-wave, and $|\cos(2\varphi)|$ for a nodal gap.

**DFT calculations of the electronic band structure.** We used van der Waals density (vdW) functional and the projector-augmented wave (PAW) method[62], as implemented in the VASP code[63]. We adopted the generalized gradient approximation (GGA) proposed by Perdew et al. (PBE)[64] and DFT-D2 vdW functional proposed by Grimme et al.[65] as a nonlocal correlation. Spin–orbit coupling (SOC) is included in all cases. A plane wave basis with a kinetic energy cutoff of 500 eV was employed. We used a $\Gamma$-centered **k**-point mesh of $15 \times 9 \times 5$. Optimized lattice parameters of $T_d$ phase are $a = 3.507$, $b = 6.371$, and $c = 13.743$ Å, close to the previous experimental values; $(a, b, c) = (3.468, 6.310, 13.861)$[8] and $(3.458, 6.304, 13.859)$[3].

**Data availability.** All relevant data are available from the authors. The data can also be found at the following link http://musruser.psi.ch/cgi-bin/SearchDB.cgi using the details: GPD, Year: 2016, Run Title: MoTe$_2$.

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

## Acknowledgements

The μSR experiments were carried out at the Swiss Muon Source (SμS) Paul Scherrer Insitute, Villigen, Switzerland. X-ray PDF measurements were conducted on beamline 28-ID-2 of the National Synchrotron Light Source II, a U.S. Department of Energy (DOE) Office of Science User Facility operated for the DOE Office of Science by Brookhaven National Laboratory under Contract No. DE-SC0012704. Z.G. gratefully acknowledges the financial support by the Swiss National Science Foundation (Early Postdoc Mobility SNFfellowship P2ZHP2-161980). The material preparation at

Princeton was supported by the Gordon and Betty Moore Foundation EPiQS initiative, Grant GBMF-4412. Z.G. and Y.J.U. thank Andrew Millis and Rafael Fernandes for useful discussions. Work at Department of Physics of Columbia University is supported by US NSF DMR-1436095 (DMREF) and NSF DMR-1610633 as well as REIMEI project of Japan Atomic Energy Agency. A.N.P. acknowledges support from the US National Science foundation via grant DMR-1610110. Work in the Billinge group was supported by U.S. Department of Energy, Office of Science, Office of Basic Energy Sciences (DOE-BES) under contract No. DE-SC00112704. S.B. acknowledges support from the National Defense Science and Engineering Graduate Fellowship program. A.S. acknowledges support from the SCOPES Grant No. SCOPES IZ74Z0-160484. B.A.F. achknowledges support from DOE-BES Materials Sciences and Engineering Division under Contract No. DE-AC02-05-CH11231 and Grant No. DE-AC03-76SF008. CAM and AL were supported by the NSF MRSEC program through Columbia in the Center for Precision Assembly of Superstratic and Superatomic Solids (DMR-1420634). Additionally, this research used resources of the National Energy Research Scientific Computing Center, a DOE Office of Science User Facility supported by the Office of Science of the U.S. Department of Energy under Contract No. DE-AC02-05CH11231.

## Author contributions

Project planning: Z.G.; Sample growth: F.v.R. and R.J.C.; μSR experiments: Z.G.; Z.S.; R.K.; A.A.; H.L.; C.B.; E.M.; A.S.; G.T.; B.F., Z.G. and Y.J.U.; μSR data analysis: Z.G.; data interpretation: Z.G., A.R.W., A.N.P. and Y.J.U.; X-ray pair distribution function measurements and analysis: S.B., Z.G. and S.Bi.; DFT calculations: A.T.L. and C.A.M.; Draft writing: Z.G. with contributions and/or comments from all authors.

## Additional information

**Competing interests:** The authors declare no competing financial interests.

**Change history:** A correction to this article has been published and is linked from the HTML version of this paper.

