## [Peer Review File · Nature Communications]

Reviewers' comments:

Reviewer #1 (Remarks to the Author):

This paper makes a convincing case, using μ SR experiments, that the superconducting state in MoTe₂ has a two-gap structure and the authors study this as a function of pressure (in fact pressure is needed to realise this state). The research has been carried out well and the paper makes its case clearly. The demonstration of a topologically non-trivial $s^{\{+-\}}$ state is somewhat circumstantial (maybe this conclusion is over-argued), but the authors make a good case for arguing it to be likely.

Also of great interest is the persistence of Uemura scaling that is observed using pressure as a tuning parameter - remarkable that this is seen even though the superfluid stiffness is enhanced (or T_c is reduced) in this family by a factor of ~ 10 from the main scaling line (putting this material further to the BCS end of the BCS-BEC crossover).

These results are of great interest and I support the publication of this paper in Nature Communications.

I only have one minor comment which is that the writing in the paper could be improved. For example, the title is missing a definite article. (One is also missing from before the first $s^{\{+-\}}$ in the abstract.) The manuscript would benefit from a light stylistic revision of the English throughout. Also, the use of the term "s+s-wave" makes it look like a combination of s+ and s-. In fact, I think the "+" is a plus, but the "-" is a hyphen. The authors sometimes use brackets, as in "(s+s)-wave", and I think this is much better, but it is not used consistently (e.g. in Figure 4). A consistent use of this would be better.

Reviewer #2 (Remarks to the Author):

This is a paper reporting some interesting and probably important results but it is not very well written in places and needs revision before it can be accepted.

I don't think the value of magnetic field used for the TF μ SR measurements is recorded anywhere. I would guess that it is larger than that given by the apparent frequency of the oscillating signals which are probably translated into a rotating reference frame. In any case, the value needs to be stated in order that the distorting effects of being near H_{c1} or H_{c2} can be ruled out.

page 3 top left: the reference [15] giving p-dependent T_c & H_{c2} should be quoted at this point.

page 4 bottom left. I am not convinced that the noise is small enough to state that there is an upturn at 700 mK, although I am happy that a 2-gap model is required.

If I understood the sample description, the μ SR sample was a polycrystalline assembly of randomly oriented crystallites. If that is the case, Eqn 1 is not correct: it applies for the field applied perpendicular to the planes. For a fairly anisotropic sample, as expected here, randomly oriented, the value of λ is given by Barford & Gunn Physica C 156 515-22 (1988) (this is a 23% correction)

The two-gap model is described as the alpha-model, but then in Eqn (2) weights w_1 & w_2 are used rather than α & $(1-\alpha)$

Also there is no mention in the main text of the temperature-dependence of Δ (BCS-like). Indeed, Eqn (2) taken literally, implies that there is no temperature-dependence and Δ is

constant at its low temperature value.

1- & 2-gap s models are referred to as s-wave And s + s-wave models. It is confusing to have two names for the same thing, and there is a possibility of the reader also assuming the dashes are somehow related to s+/s- superconductivity, which is not implied at this point. I would prefer 1-gap s and two-gap s. Finally, the nodal model should be referred to as nodal, rather than d-wave, since we are dealing with an orthorhombic material. This applies to text and to Fig 4. In Fig 4 (a) there is a legend data0, which should be removed.

Why in the fits was the ratio of weights fixed?

The weighted value of $\Delta(0)$ should be compared with the BCS value expected from T_c . I think it will be quite close. This would provide a useful corrective to the overemphasis of the Uemura plot discussed at length at the bottom of page 6 and top of page 7.

I feel that the conclusion that this material is a topological superconductor, with different signs of Δ at the different nodes is justified but not strong

Reply to Referee A

We are grateful to Referee A for supporting the publication of the paper in Nature Communications and for giving useful comments. We have modified the manuscript accordingly:

1. *This paper makes a convincing case, using μ SR experiments, that the superconducting state in MoTe₂ has a two-gap structure and the authors study this as a function of pressure (in fact pressure is needed to realise this state). The research has been carried out well and the paper makes its case clearly. The demonstration of a topologically non-trivial s^{+-} state is somewhat circumstantial (maybe this conclusion is over-argued), but the authors make a good case for arguing it to be likely.*

Also of great interest is the persistence of Uemura scaling that is observed using pressure as a tuning parameter - remarkable that this is seen even though the superfluid stiffness is enhanced (or T_c is reduced) in this family by a factor of 10 from the main scaling line (putting this material further to the BCS end of the BCS-BEC crossover).

These results are of great interest and I support the publication of this paper in Nature Communications.

I only have one minor comment which is that the writing in the paper could be improved. For example, the title is missing a definite article. (One is also missing from before the first s^{+-} in the abstract.) The manuscript would benefit from a light stylistic revision of the English throughout. Also, the use of the term "s+s-wave" makes it look like a combination of s_+ and s_- . In fact, I think the "+" is a plus, but the "-" is a hyphen. The authors sometimes use brackets, as in "(s+s)-wave", and I think this is much better, but it is not used consistently (e.g. in Figure 4). A consistent use of this would be better.

According to the suggestion of the Referee A, a stylistic revision of the manuscript was carefully done. In addition, we consistently use the definitions "1-gap s-wave and 2-gap s-wave" throughout the revised manuscript.

Reply to Referee B

We would like to thank referee B very much for carefully reading our manuscript and finding our results interesting. We have modified the manuscript accordingly and in the following we will respond to the points raised by him / her:

1) *This is a paper reporting some interesting and probably important results but it is not very well written in places and needs revision before it can be accepted.*

I don't think the value of magnetic field used for the TF μ SR measurements is recorded anywhere. I would guess that it is larger than that given by the apparent frequency of the oscillating signals which are probably translated into a rotating reference frame. In any case, the value needs to be stated in order that the distorting effects of being near H_{c1} or H_{c2} can be ruled out.

Transverse field μ SR experiments were performed in an applied magnetic field of $\mu_0 H = 20$ mT. The value of the field is larger than $H_{c1} \sim 5$ mT but much smaller than the second critical field H_{c2} . This makes sure that the separation between the vortices is smaller than λ and allows a reliable determination of the penetration depth. We mention the value of magnetic field at various relevant places in the revised manuscript.

page 3 top left: the reference [15] giving p -dependent T_c and H_{c2} should be quoted at this point.

We considered this suggestion of the Referee B in the revised manuscript.

page 4 bottom left. I am not convinced that the noise is small enough to state that there is an upturn at 700 mK, although I am happy that a 2-gap model is required.

We agree with the Referee that the upturn is not very well pronounced and without analysing the T -dependence of the penetration depth data, it is difficult to conclude on the existence of multiple SC gaps. Thus, the sentence

stating the upturn at 700 mK is replaced by more general sentence, which directs the reader to the next section, where the presence of the two isotropic s -wave gaps on the Fermi surface of MoTe_2 is justified by precise analysis.

If I understood the sample description, the μSR sample was a polycrystalline assembly of randomly oriented crystallites. If that is the case, Eqn 1 should be revised: it applies for the field applied perpendicular to the planes. For a fairly anisotropic sample, as expected here, randomly oriented, the value of lambda is given by Barford and Gunn Physica C 156 515-22 (1988) (this is a 23 % correction)

Regarding the penetration depth, it is the effective penetration depth λ_{eff} (powder average) which we extract from the μSR depolarization rate (Eqn. 1) and this is the one shown in the figures. It was shown [Physica C 156, 515-22 (1988) and Physica C 176, 551-558 (1991)] that in polycrystalline samples of highly anisotropic systems λ_{eff} is dominated by the shorter penetration depth λ_{ab} . According to Eq. 42 of the reference Physica C 176, 551-558 (1991), $\lambda_{eff} = 1.3\lambda_{ab}$. Note that lower factor between λ_{eff} and λ_{ab} was reported in the reference Physica C 156, 515-22 (1988), but it was corrected later in Physica C 176, 551-558 (1991) and the factor 1.3 is the one which is used in all the previous μSR papers on polycrystalline samples. In the paper, we decided to show λ_{eff} and not λ_{ab} . Also in the Uemura plot, the values of $\lambda_{eff}^{-2}(0)$ are shown. In the revised manuscript, we make this point clear and explicitly mention which penetration depth we refer to. In Eqn. (1,2) and also in Figs. (4,5) λ is replaced by λ_{eff} .

The two-gap model is described as the alpha-model, but then in Eqn (2) weights w_1 and w_2 are used rather than alpha and (1-alpha)

This point of Referee B is considered in the revised manuscript. Specifically, we replace ω_1 and ω_2 by α and $(1-\alpha)$, respectively.

Also there is no mention in the main text of the temperature-dependence of Delta (BCS-like). Indeed, Eqn (2) taken literally, implies that there is no temperature-dependence and Delta is constant at its low temperature value.

In the penetration depth analysis, the temperature and the angular dependence of the gap is given by: $\Delta_i(T, \varphi) = \Delta_{0,i}\Gamma(T/T_c)g(\varphi)$ ($\Delta_{0,i}$ is the

maximum gap value at $T = 0$), where the temperature dependence of the gap is approximated by the expression $\Gamma(T/T_c) = \tanh \{1.82[1.018(T_c/T - 1)]^{0.51}\}$, while $g(\varphi)$ describes the angular dependence of the gap. Due to the page limit, all the above details are given in the methods section. In the revised version of the main text, the corresponding sentence, referring to the methods section, is added.

1- and 2-gap s models are referred to as s-wave And s + s-wave models. It is confusing to have two names for the same thing, and there is a possibility of the reader also assuming the dashes are somehow related to s+/s-superconductivity, which is not implied at this point. I would prefer 1-gap s and two-gap s. Finally, the nodal model should be referred to as nodal, rather than d-wave, since we are dealing with an orthorhombic material. This applies to text and to Fig 4. In Fig 4 (a) there is a legend data0, which should be removed.

According to the suggestion of the Referee B, we consistently use the definitions “1-gap s-wave and 2-gap s-wave“ throughout the revised manuscript.

Why in the fits was the ratio of weights fixed?

From the initial analysis we realised that the ratio of weights do not change under pressure within the error bars. Thus, in the final step of the analysis we decided to fix the ratio in order to reduce the number of free parameters during the fitting.

The weighted value of Delta(0) should be compared with the BCS value expected from T_c . I think it will be quite close. This would provide a useful corrective to the overemphasis of the Uemura plot discussed at length at the bottom of page 6 and top of page 7.

This is an interesting of the Referee B. The gap to T_c ratios were estimated to be $2\Delta_1/k_B T_c = 1.5(4)$ and $2\Delta_2/k_B T_c = 4.6(5)$ at $p = 0.45$ GPa for MoTe_2 , for the small and the large gaps, respectively. The ratio for the higher gap is consistent with the strong coupling limit BCS expectation. However, similar ratio can also be expected for Bose Einstein Condensation (BEC)-like picture as it was previously pointed out [Nature Materials **8**, 253 (2009)]. It is important to note that the ratio $2\Delta/k_B T_c$ does not allow distinguishing

BCS or BEC condensation. Particularly in two band systems, where the ratio is not universal even in the BCS limit, as it depends also on the density of states of the two bands. We add this discussion in the revised manuscript and give the corresponding references.

I feel that the conclusion that this material is a topological superconductor, with different signs of Δ at the different nodes is justified but not strong

In this respect, we think that our manuscript is already very balanced.

REVIEWERS' COMMENTS:

Reviewer #1 (Remarks to the Author):

The paper has been revised according to the comments from both referees and both the response and the revised manuscript address the points raised. I support publication.

Reviewer #2 (Remarks to the Author):

I am happy with the revised MS, except for one stylistic matter:

In the abstract and in the text the authors write that the superconductivity in this material is:

"... determined to be 2-gap s-wave symmetric"

I think that a better way of writing this is:

"... determined to have 2-gap s-wave symmetry"

Manuscript NCOMMS-17-10697A

Reply to Reviewer 1

The paper has been revised according to the comments from both referees and both the response and the revised manuscript address the points raised. I support publication.

We are very grateful to Reviewer 1 for supporting the publication of the paper in Nature Communications.

Reply to Reviewer 2

I am happy with the revised MS, except for one stylistic matter: In the abstract and in the text the authors write that the superconductivity in this material is: "... determined to be 2-gap s-wave symmetric" I think that a better way of writing this is: "... determined to have 2-gap s-wave symmetry".

We are very grateful to Reviewer 2 for the positive and supportive report and for his/her useful suggestion. According to the suggestion of the Reviewer 2, the mentioned one stylistic matter was considered. Specifically, we replaced "... determined to be 2-gap s-wave symmetric" by "... determined to have 2-gap s-wave symmetry" throughout the revised manuscript.